# Reuse of intermittent catheters: a qualitative study of IC users' perspectives

Miriam Avery,[1] Jacqui Prieto,[1] Ikumi Okamoto,[1] Samantha Cullen,[1] Bridget Clancy,[1] Katherine N Moore,[2] Margaret Macaulay,[1] Mandy Fader[1]

¹School of Health Sciences, University of Southampton, Southampton, UK
²Faculty of Nursing, University of Alberta, Edmonton, Alberta, Canada

**Correspondence to**
Dr Miriam Avery;
M.Avery@soton.ac.uk

## ABSTRACT

**Objectives** To explore the views of intermittent catheter (IC) users regarding the advantages and disadvantages of single-use or reuse of catheters.

**Design** Qualitative study with semi-structured interviews. The interviews were recorded, transcribed and analysed thematically.

**Setting** Participant's own homes in Hampshire and Dorset, UK.

**Participants** A convenience sample of 39 IC users, aged 23–86 years, using IC for at least 3 months.

**Results** The analysis revealed four main themes: concerns regarding risk of urinary tract infection (UTI); cleaning, preparation and storage; social responsibility; practicalities and location. The main concern was safety, with the fear that reuse could increase risk of UTI compared with single-use sterile catheters. If shown to be safe then around half of participants thought they might consider reusing catheters. The practicalities of cleaning methods (extra products, time and storage) were considered potentially burdensome for reuse; but for single-use, ease of use and instant usability were advantages. Always having a catheter without fear of 'running out' was considered an advantage of reuse. Some participants were concerned about environmental impact (waste) and cost of single-use catheters. The potential for reuse was usually dependent on location. The analysis showed that often the disadvantages of single-use could be off-set by the advantages of reuse and vice versa, for example, the need to take many single-use catheters on holiday could be addressed by reuse, while the burden of cleaning would be obviated by single-use.

**Conclusions** If shown to be safe with a practical cleaning method, some participants would find reuse an acceptable option, alongside their current single-use method. The choice to use a mixture of single-use and reuse of catheters for different activities (at home, work or holiday) could optimise the perceived advantages and disadvantages of both. The safety and acceptability of such an approach would require testing in a clinical trial.

## INTRODUCTION

Intermittent catheterisation (IC) is well-established management for children and adults with chronic urinary retention. The positive benefits of IC are well-known; IC users have reported relief of symptoms

of frequency, urgency and incontinence, improved sleep and fewer restrictions on daily physical activities[1 2] and improved quality of life (QOL).[3] Although the value of IC is clear, some users describe the challenges they encounter. They report negative effects on QOL, both psychologically and socially, and urinary tract infection (UTI) remains a common problem.[1 2] In one cross-sectional survey study, only half of the 44 participants were completely satisfied with IC.[4]

Research about IC-related issues, in particular UTI, has focused on single-use catheters (the catheter is used once then discarded) with the development of new catheter coatings, prelubrication and compact designs. Reuse of uncoated catheters (the same catheter cleaned and reused by one individual several times) has gained little attention from researchers or industry, even though single-use catheters may not be the most environmentally friendly or cost-effective strategy. Catheter reuse continues in some developed countries, but is now rare in the UK since the introduction of single-use coated catheters and subsequent changes in medical device legislation. An international guideline, prepared by the Infectious Diseases Society of America (IDSA)[5] concluded that for those living in the community and using

clean self-catheterisation, the differences between sterile catheters (single-use) or reused catheters are not significant and that there is no convincing evidence that one catheter design or method is better than another in terms of UTI. This leaves clinicians with insufficient information to recommend one approach over the other.

A review of the literature was conducted and four papers were found which, to varying degrees, included user experiences of reusing IC catheters compared with single-use.[6–9] Two were qualitative studies exploring individuals' experiences of IC in general; issues raised included cost and the environmental impact of single-use compared with reuse, from a UK and US perspective.[6 7] In two other studies,[8 9] children with spina bifida and their families were asked about their experiences of using a new hydrophilic-coated single-use catheter, compared with their usual method of reusing an uncoated catheter. Some described the single-use catheter as more hygienic and felt that it was 'safer' for their child.[9] Single-use was also rated positively regarding convenience and for outings where the toilet facilities were not satisfactory.[8 9] However, the packaging and the environmental impact of single-use catheters were raised as downsides of single-use catheters.[9]

Although the potential benefits of reuse may appeal to some IC users, the potential risks and challenges are not well understood. This study is part of the National Institute for Health Research funded Mult*I-C*ath programme,[10] which aims to develop and test a catheter cleaning method using laboratory techniques and patient panels as well as using interviews and surveys to determine IC user and clinician perspectives[11] regarding reuse of catheters. The purpose of this study is to explore the views of community dwelling IC users regarding the advantages and disadvantages of single-use or reuse of uncoated catheters.

## METHOD

This study used a qualitative in-depth interview method. Findings from a subset of these interviews, related to the views of IC users about UTI symptoms and management, have recently been published.[12] A convenience sample of 39 IC users was recruited from 12 general practitioner (GP) practices in Hampshire and Dorset, UK. Potential participants were identified via a computerised database search using prescription codes. GPs checked eligibility (box 1) and an invitation letter with a reply form, Participant Information Sheet and FREEPOST envelope were sent out by practice staff. At 3 weeks, reminders were sent to non-responders. Recruitment for the study continued until it was deemed data saturation had been reached. Informed written consent was obtained from each participant and all information was kept confidential.

---

**Box 1    Inclusion and exclusion criteria**

**Inclusion criteria**
► Over 18 years
► Currently using intermittent catheters (IC) for 3 months or more

**Exclusion criteria**
► Urethral stricture or deformity
► Immune deficiency disorder (increased infection risk)
► External carer required for IC (eg visiting community nurse performs IC).

---

### Data collection

All interviews were conducted face-to-face in participants' own homes by one of three trained female interviewers, including a postdoctoral qualitative researcher (SC) and two research nurses, each of whom had at least 10 years' research experience (BC and MA). None of the interviewers had been in previous contact with any of the participants and they attended in a non-clinical capacity as non-judgemental listeners. A relative or carer was present for the interview if requested by the participant. A semi-structured interview schedule was used to explore experiences, values and beliefs about IC (see online supplementary file 1). Each interview lasted approximately 45–60 min. Interviews were recorded using digital audio recording equipment and then transcribed verbatim. All transcripts were anonymised with an assigned study number. The transcripts were read only by the researchers and not by participants. Field notes were made during interviews, which were used in the analysis.

### Data analysis

Data from interviews and field notes were analysed thematically using the method described by Braun and Clarke[13] and coded using NVivo10 (QSR International) by a qualitative researcher (IO) who had not conducted the interviews. This involved moving back and forth between interview transcripts and field notes, with reference to the research literature. An initial coding framework was derived from the first 20 interview transcripts and from this the main themes emerged and subthemes were developed. These were further refined through discussion with the wider research team and ongoing analysis of the remaining transcripts. Any newly emerged themes that did not fit into the initial themes were discussed as a team. No participants were involved in the analysis of data.

### Patient and public involvement

The research question was derived from a consumer-clinician priority setting exercise by the James Lind Alliance in 2008. Patient and Public Involvement representatives (two women and one man) were members of the Mult*I-C*ath Project Management Group and supported the programme to ensure that participant experience was considered at each stage of the research. They contributed to the development of the study design, processes

and materials. Bladder and Bowel UK, a key incontinence consumer organisation, is a collaborator on the Mult*I-C*ath programme.

## RESULTS
### Participants
A total of 139 IC users were invited to take part. Seventy-four (53%) responded, of whom 42 were willing to participate. Three were ineligible, so 39 were included. The participants included 24 men and 15 women, with a mean age of 67 years (range 23–86 years) who had been using IC for a mean length of 8.5 years (range 9 months–30 years). The frequency of IC ranged from once every second day to 10 catheterisations a day (mean 4/day). Reasons for IC included chronic urinary retention (n=20), neurological impairment (n=10) and other factors such as surgery or chemotherapy (n=9).

The majority of participants (n=34) were using a catheter once and discarding it after use. Most (n=30) were using hydrophilic-coated catheters, of whom 10 used catheters with a compact design. Of the remaining five, four were currently reusing catheters designed for single-use on an ongoing basis, and one had previously reused catheters, switching to single-use 10 years ago because of convenience. The four participants who reused catheters were all men who had been reusing for between 10 and 30 years. Their reasons for reuse were concerns about waste or cost.

### Qualitative findings
The results focus on the perspectives of the participants regarding single-use and reuse of IC catheters; themes emerged about safety, burden and lifestyle. The four main themes were concerns regarding risk of UTI; cleaning,

preparation and storage; social responsibility and practicalities and location. The themes and subthemes are described below and shown in figure 1. Those who had never reused catheters were asked about their views but not their own experiences.

Overall, around half of the single-use participants indicated they would not want to reuse IC catheters or had major concerns about whether it would be an acceptable method. For the remaining participants, the potential acceptability of reuse was very much dependent on the safety and efficacy of the cleaning regimen and whether there was an increased risk of infection, as well as factors such as convenience and practicalities (figure 1).

### Concerns regarding risk of UTI
*Safety: fears about increasing risk of UTI*
The question of UTI risk was the main concern raised by many single-users when asked about reusing catheters. This was the case for those who had suffered UTIs and others who had not and for men and women participants. They described their concerns about infection as their biggest worry and for some it was the only issue they had when considering reuse.

> Infection (concerns me about re-using catheters) definitely. Especially as I've not had any infection, I think I'm doing something right, if you get me. (Participant 22, female, single-user)

> My honest opinion (about reusing) is I can understand why they're trying to do it, it's they're trying to save money. But in my opinion there's always a chance that that's going to go wrong and I'm the one who's gonna end up having a water infection or something like that. (Participant 13, male, single-user)

| MAIN THEME | SUB-THEME | CODES | | | |
|---|---|---|---|---|---|
| | | RE-USE | | SINGLE-USE | |
| | | Advantages | Disadvantages | Advantages | Disadvantages |
| 1. Concerns regarding risk of UTI | 1.1 Safety: fears about increasing risk of UTI | | Concern about hygiene and perceived link to UTI | Considered less likely to cause infection Sterile | |
| | 1.2 Necessity of safe cleaning | | Cleaning and preparation needed | | |
| 2. Cleaning, preparation and storage | 2.1 Concerns about a complicated, time-consuming cleaning procedure | | Cleaning and preparation needed | | |
| | 2.2 Concerns about storage/supply of catheters | Always having one No storage problem | Storing equipment Concern use-by-date could expire | Good delivery system/supplier | Shortage of supply/not good delivery service |
| 3. Social responsibility | 3.1 Reducing costs to National Health Service | Cost | | | Cost |
| | 3.2 Reducing waste/impact on environment | Less impact on the environment | | | Environment and waste |
| 4. Practicalities and location | 4.1 Freedom and a normal life | | Limiting lifestyle | Instantly usable or easy to use and to dispose No pain and easy to use Gives freedom/normal life More independence | |
| | 4.2 Travel | Less to carry when travelling | Taking cleaning equipment on holiday | | Not easy to carry on holiday |
| | 4.3 Carrying used catheters | | Carrying used catheters | | |
| | 4.4 Discretion | | Not discreet | Easy to carry and discreet Ready to use Easy to use | Don't like to dispose used catheters in public toilets Not easy to carry around |
| | 4.5 Choice of catheters – mixed use | Emerged from interviews | | | |

**Figure 1** Themes, subthemes and coding from analysis of qualitative interviews. UTI, urinary tract infection.

Many perceived that using single-use catheters meant they had less chance of getting an infection and they felt that the single-use method was more 'clean' and 'sterile'.

(Single-use catheters have) less chance on (of) infection, you can dispose of it, get rid of it. (Participant 40, male, single-user)

However, the four participants who were reusing were not deterred by fears about infection.

### Necessity of safe cleaning method

Related to the fears described above, some single-users described the importance of an effective catheter cleaning method as a prerequisite for reuse.

(It is important) that you follow all the steps that you are supposed to. Because if you don't then it might not be sterile. Because I don't really know what it, what those effective ways would be. (Participant 16, female, single-user)

### Cleaning, preparation and storage
### Concerns about a complicated/time-consuming cleaning procedure

Many single-users thought that the cleaning procedure would be complicated and were concerned about the amount of work and time involved. Some did not want to be bothered with it and thought it could be inconvenient and potentially messy.

….but I mean if you had to go to all the bother of washing a catheter, I don't know how you'd do it to be quite honest, to get it really sterile. (Participant 29, female, single-user)

Some single-users mentioned that they needed to know more about the cleaning method before they could comment on reuse. It appeared, however, that all four of the catheter reusers were conducting a very simple catheter cleaning procedure, typically using only tap water to rinse out used catheters.

Holding it under the tap (running it under tap water), and then I put it right back in the pouch, and then it's as clean as a whistle. (Participant 19, male, reuser)

### Concerns about storage and supply of catheters

The participants who would consider re-use of catheters thought it could save space in the home. However, some suggested that finding space to store extra equipment for cleaning catheters would be a problem. One participant, with a complex long-term health condition that required other equipment, was concerned about this due to the amount of equipment she already had.

My bathroom is already clogged full of all my bulky stuff, I already have my sort of like bag that hangs up for flushing my bowels out.(…) I think that I'd want it to not be too bulky. (Participant 11, female, single-user)

In terms of catheter supply, although some participants were very happy with their current supply arrangements, for others the prospect of a supply shortage was a drawback of single-use catheters. Some participants thought they would not need to worry about running out if they could reuse them.

Wouldn't have to keep running up the chemist (if I used reusable catheters). You'd know you've always got one. You would know there's always one there. (Participant 7, female, single-user)

### Social responsibility
### Reducing costs to NHS

Many participants expressed their concerns or could see the issues around the financial cost of their catheters for the National Health Service (NHS) and some suggested that they would consider reuse to contribute to reducing such cost.

I think the GP's surgery pays for (my catheter) but I found out how much it was last week and I was shocked, really shocked actually. I feel guilty about that. (Participant 15, female, single-user)

Some described the potential cost-saving as the only benefit of reuse and noted that the benefit was not to them directly. Two participants questioned whether there would be any saving, with one considering that it may cost more due to the potential for increased infections.

### Reducing waste and the impact on the environment

Several participants felt that reusing catheters would reduce waste and, for that reason, might be more environmentally friendly.

The good thing about re-using would be that it would cut down the amount of waste. (…) If I go through 240 (single-use catheters) every 6 weeks that is a mountain of catheters every year I get through. So I think that is the main benefit really. (Participant 38, female, single-user)

A participant who was reusing indicated that not disposing of the catheter after each use alleviated the worry about contributing to landfill waste. Another reuser described how reducing waste was the main motive to start reusing their catheters.

I hate wasting anything, and so that's all built into why I use the catheter more than once. (Participant 25, male, reuser)

### Practicalities and location
### Freedom and a normal life

Most single-users appreciated their current catheters as they were convenient, comfortable and easy to use, with some describing them as giving freedom and a normal life; IC had become part of life, a natural thing like cleaning teeth.

I love that (my catheter) is compact. (…) They are just so convenient to use. No fuss. It doesn't impact on your normal daily lifestyle. They are great for travelling. That's what I love about my catheters. And it gives me a normal life; otherwise I would probably have to be at home more really. (Participant 38, female, single-user)

The participants were often familiar with the design of their current single-use catheter and described how it suited them personally, although this may not directly relate to single-use or reusability.

When considering reuse, the participants discussed the locations where they would be more likely to reuse their catheters; they often compared the practicalities of reuse inside and outside the home as well as issues around travel.

### *Travel*
The compact size of single-use catheters was appreciated by participants because they were easy to carry individually. However, for holidays, the need to take a large supply was a hindrance. In this situation, participants noted that reuse of catheters could be a more practical option.

One participant valued this aspect of reusing from his experience.

If I take (my catheters), I'll take 10 for the month or something. If I take the other one I've got to take 120. (…) I did a trip around Australia once with my family and we were away 30 days or a month and the difference would have been huge. I pretty much would have needed a suitcase just for that. (…) So that's always been a major issue. (Participant 37, male, reuser)

However, several single-users were concerned about carrying the cleaning equipment on holiday.

### *Carrying used catheters*
Many participants appreciated the portability and ease of disposal of the single-use catheters. Some single-users, therefore, wondered what they would have to do with the used catheters if they catheterised when they were out.

If you are going out for the day. You'd use your catheter in the morning; I don't want to take that one around with me. I don't know how I could. (Participant 41, male, single-user)

However, one of the reusers emphasised the possible flexibility of reusing their catheter, as they could also be thrown away after a single use if needed.

You can always throw it away and you've only done the same as you've done with a non-re-usable one. So if you find yourself out and about and it's not as hygienic as you'd have liked; (…) I will tend to use it once and throw it away under those circumstances. (Participant 37, male, 50, single-user)

### Discretion
The majority of single-users appreciated the compact packaging and discretion of their current catheters. Many were concerned they would not have these advantages if they reused catheters and the lack of discretion would restrict them.

(Single-use catheters are) just quick really, it doesn't take me any longer than somebody else going into the toilet, not much longer at all so it's that easy. If you have to start cleaning it after using it….while you're out, it then gets a bit (much). And it's obviously difficult to do it in a public toilet as well. (Participant 3, female, single-user)

While some said that they would be happy to reuse at home, others expressed their concern, as they did not want visitors to see used catheters or the cleaning equipment.

### Choice of catheters—mixed use
Among those who viewed reuse as potentially acceptable, most said they would value having both types of catheters to choose from depending on the situation.

Important stuff would be that when I'm doing it at home it's not generally a problem. If I know what I've got to do I will do it. It's knowing that when you are going away, you don't know where you are going to, (…) is that equipment there? For me to be able to do it? (Participant 1, male, single-user)

…..I wouldn't want to be forced to have just re-used catheters. But in most circumstances I would choose to use it except for when I'm travelling in which case I would definitely want to be using the disposable ones. And I think generally speaking if you are away you don't want all the faff. (Participant 38, female, single-user)

## DISCUSSION
We believe this is the first study to explore IC users' perspectives on the advantages and disadvantages of single-use and reuse of intermittent catheters. The findings highlight the complexity of factors involved and show the variation of views among IC users; for some, reuse is not an acceptable option and for others it could be acceptable, but robust evidence is needed regarding safety, and the cleaning method must be shown to be effective and user-friendly.

### Concerns regarding risk of UTI
For IC users to consider reuse they must first be satisfied that it is safe to do so; this was the crucial factor for many. In fact, around half of participants in this study said they would not consider reuse or had major concerns about it, mainly due to fears about infection. A guideline for the diagnosis, treatment and prevention of catheter-associated UTI (IDSA) concluded that

there have not been enough robust studies to provide evidence regarding risk of UTI when comparing reuse of catheters with single-use catheters.[5] This emphasises the need for a randomised controlled trial to properly examine this.

### Cleaning, preparation and storage

The development of a cleaning method that is practical for IC users was an issue raised by participants. Concerns were voiced about the number of cleaning items requiring storage and the potential burden of doing the cleaning as well as the effectiveness of the cleaning method. Others have suggested that the cleaning method could be a burden when reusing catheters,[14] which could in turn impair both independence and adherence.[15] Chick et al found that some families in their study preferred the convenience of the one-step process for single-use catheters.[8] It is therefore important that the cleaning method is developed with IC users alongside rigorous laboratory testing.[16]

### Social responsibility

Despite participants' concerns about reuse, they recognised some advantages. Many thought there could be cost savings for the NHS and a positive environmental impact and some were currently reusing, or would consider reusing, for these reasons. Similar findings were reported by Kelly et al[6] in which 16 individuals were interviewed about their experiences and concerns about IC use. Financial burden to the health service and environmental impact were raised by 63% (10/16) and 38% (6/16) of the participants, respectively; the authors suggested this reflects the greater awareness of environmental issues in society.[6] Although several of the participants in this study felt that reuse of catheters would be more environmentally friendly, this is a complex issue dependent on the method used for cleaning; there is a need for knowledge to be gained on environmental issues.

Four distinct modelling studies on cost-effectiveness using different IC catheters and methods have been published. Both Neovius et al[17] and Bermingham et al[18] compared reuse with single-use catheters. In a conference abstract, Neovius et al concluded that reuse of PVC catheters was not cost-effective compared with single-use coated catheters,[17] whereas Bermingham et al indicated that reuse could be more cost-effective than single-use catheters but that additional evidence was required before any practice changes could be recommended.[18] Clark et al[19] concluded that single-use coated rather than single-use uncoated catheters were more cost-effective when outcomes were calculated over a patient's lifetime. Neovius and Lundqvist[20] indicated that patients would be willing to pay out of pocket for a hypothetical catheter that would reduce/prevent UTIs. The probabilistic decision model of these studies regarding risk of UTI limits the interpretation of the results and indicates, as Bermingham et al stated, that further robust evidence is required.

### Practicalities and location

Although many of the participants would consider catheter reuse, willingness to do so varied between individuals based on lifestyle and circumstances. The need for discretion was of great concern and was closely linked to location; some participants would only consider reusing catheters when they were at home. The literature also confirms that location, and specifically whether at home or away from home, has a bearing on how people feel about reuse.[8 9] This highlights the need for a flexible approach to IC methods, which may be achieved by giving IC users the option to use both catheter types to suit their lifestyle and preferences.

Many of the perceived disadvantages of reuse are off-set by advantages of single-use and vice versa (figure 1). For example, some IC users were concerned about running out of their single-use catheters, and the ability to reuse their catheter would mean they always have a usable catheter. Others emphasised the benefit of using single-use catheters when in public places and the difficulties of reuse when away from home reinforcing the potential benefit of mixed use. Furthermore, if reuse of catheters was more commonplace, efforts could be made to design and manufacture discreet, reusable versions in the same way as for single-use.

### Limitations

As the primary aim of the study was to explore whether reusing catheters would be acceptable to IC users in preparation for a randomised controlled trial, and this was stated in the participant information sheet, we may have recruited primarily individuals who were interested in issues around reuse. Therefore, findings may not be representative of all UK IC users. Furthermore, the perspectives of single-use IC users about reusing catheters were hypothetical and not based on their actual experiences. The inclusion of questions about advantages and disadvantages of both methods may have influenced participant's responses and the researcher's presence during the data collection might have elicited socially desirable responses; however, the researcher presented as an impartial observer during the data collection and every effort was made to keep the interviews balanced.

### Implications for future research/clinicians/policy makers

This study has shown that some IC users would find reuse of catheters acceptable, but there is a need to develop a rigorous cleaning method for IC catheters which is effective and user-friendly. If shown to be safe in a clinical trial, then efforts could be made to develop user-friendly catheter designs, for example, discreet, compact designs with an efficient cleaning method.

### CONCLUSION

This study revealed that if shown to be safe with a practical cleaning method, some participants would find reuse of catheters for some of the time an acceptable option. The

choice to use a mixture of single-use and reuse of catheters for different activities (at home, work or holiday) could optimise the advantages and disadvantages of both methods. With environmental issues and cost high on the public agenda, some IC users would advocate that if reuse is shown to be safe and acceptable then it could be made available as an option. The safety and acceptability of such an approach would require testing in a clinical trial.

**Acknowledgements** The authors would like to thank all the interviewees who agreed to take part and share their experiences.

**Contributors** MF was the chief investigator and was primarily responsible for the original grant application. MF, JP, MM, SC and MA were involved in designing the study and methods. JP led the overall study team. SC collected the majority of the data. BC and MA contributed to the data collection. IO led the analysis and IO and MA were involved in interpretation of the qualitative data and wrote draft manuscripts. IO, JP, MA, KNM, MM and MF wrote drafts and/or critically revised the manuscripts.

**Funding** This paper refers to independent research funded by the National Institute for Health Research (NIHR) under its Programme Grants for Applied Research (PGfAR) (Grant Reference Number RP-PG-0610-10078).

**Disclaimer** The views expressed in this paper are those of the authors and not necessarily those of the NHS, the NIHR or the Department of Health and Social Care (DHSC).

**Competing interests** The authors of the papers were funded by the NIHR under its Programme Grants for Applied Research (PGfAR) (Grant Reference Number RP-PG-0610-10078).

**Patient consent** Obtained.

**Ethics approval** Ethics permission was granted from an NHS Research Ethics Committee (NRES Committee London, Hampstead) (REC reference 13/LO/1511).

**Provenance and peer review** Not commissioned; externally peer reviewed.

**Data sharing statement** No additional data is available.

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
