## [Reviewer comments · BMJ Open]

ARTICLE DETAILS

TITLE (PROVISIONAL)	Re-use of intermittent catheters: A qualitative study of IC users' perspectives
AUTHORS	Avery, Miriam; Prieto, Jacqui; Okamoto, Ikumi; Cullen, Samantha; Clancy, Bridget; Moore, Katherine; Macaulay, Margaret; Fader, Mandy

VERSION 1 – REVIEW

REVIEWER	Dianne Ramm School of Health and Social Care, University of Lincoln, England, UK
REVIEW RETURNED	03-Feb-2018

GENERAL COMMENTS	This is an extremely person-centred report focusing on a qualitative research study that seeks to establish the views of people performing intermittent catheterisation in relation to the possible use of reusable catheters. It is timely, in that there is a current and growing awareness within the general population of the negative environmental impact of disposable waste. There is therefore a pressing need to establish the risks and benefits of each approach in order that healthcare practitioners in future are able to articulate the options to their clients and also provide a robust and contemporaneous evidence base for each. This primary research study is therefore significant in representing the first explorative study into how people feel about the possibility of reusing catheters. As the findings indicate that people do wish to have a range of options available, there are clear implications for manufacturers in terms of research and development. Further studies are therefore indicated in order to identify the safest and most practical method(s) of preparing used catheters, for further use, as the authors have identified. Alternatively, new catheters might be developed that can degrade naturally on disposal. Protecting the environment is likely to become more of an issue and as this study demonstrates, already represents a particular concern for some clients in addition to the cost and their perceived 'burden' on the NHS. Introducing an element of choice might also have additional benefits in enhancing the persons' feeling of control at a time when they have experienced the loss of their ability to void naturally and when their psychological resilience might, in consequence, be low.
--

REVIEWER	Maria Hälleberg Nyman Orebro University, Sweden
REVIEW RETURNED	09-Feb-2018

GENERAL COMMENTS	Reviewers comments to authors
-------------------------------

	This is an important study on re-use of intermittent catheters from the users' perspective. 1. Is the research question or study objective clearly defined? Yes 2. Is the abstract accurate, balanced and complete? Setting should be 12 general practice clinics in Hampshire and Dorset, UK. The abstract might need to be re-written if the result section is rewritten. 3. Is the study design appropriate to answer the research question? Yes 4. Are the methods described sufficiently to allow the study to be repeated? The data analysis is not described in detail. And there is no methodological reference for the thematic analysis. Is the analysis performed in the 6 steps described by Braun and Clarke? 5. Are research ethics (e.g. participant consent, ethics approval) addressed appropriately? Yes 6. Are the outcomes clearly defined? Yes 7. If statistics are used are they appropriate and described fully? NA 8. Are the references up-to-date and appropriate? Yes 9. Do the results address the research question or objective? Yes 10. Are they presented clearly? The result section is a bit hard to follow. There are four themes and eleven sub-themes. The result seems to be based on raw data rather than analyzed material. The authors provide a lot of quotations but not so much text to describe the themes and sub-themes. This indicates that data should be analyzed further. Beside the themes and sub-themes the authors provide a figure with factors affecting acceptability of re-use – some of these factors are the same as a theme or sub-theme and some are not. This is very confusing. Also, the authors provide characteristics for the participants that re-use catheters but not for those that not re-use catheters. 11. Are the discussion and conclusions justified by the results? The result section needs to be re-written, as a result of this discussion and conclusions also might be re-written. 12. Are the study limitations discussed adequately? The authors discuss a few limitations, but they do not discuss the fact that the interview-guide include some leading questions e.g. the questions on the last page of the interview-guide. 13. Is the supplementary reporting complete (e.g. trial registration; funding details; CONSORT, STROBE or PRISMA checklist)? Funding details are provided, but no checklist e.g. COREQ 14. To the best of your knowledge is the paper free from concerns over publication ethics (e.g. plagiarism, redundant publication, undeclared conflicts of interest)? Yes 15. Is the standard of written English acceptable for publication? Yes
--	---

VERSION 1 – AUTHOR RESPONSE

Reviewer 1: Dianne Ramm

This is an extremely person-centred report focusing on a qualitative research study that seeks to establish the views of people performing intermittent catheterisation in relation to the possible use of reusable catheters. It is timely, in that there is a current and growing awareness within the general population of the negative environmental impact of disposable waste. There is therefore a pressing need to establish the risks and benefits of each approach in order that healthcare practitioners in future are able to articulate the options to their clients and also provide a robust and contemporaneous evidence base for each.

This primary research study is therefore significant in representing the first explorative study into how people feel about the possibility of reusing catheters. As the findings indicate that people do wish to have a range of options available, there are clear implications for manufacturers in terms of research and development. Further studies are therefore indicated in order to identify the safest and most practical method(s) of preparing used catheters, for further use, as the authors have identified. Alternatively, new catheters might be developed that can degrade naturally on disposal. Protecting the environment is likely to become more of an issue and as this study demonstrates, already represents a particular concern for some clients in addition to the cost and their perceived 'burden' on the NHS. Introducing an element of choice might also have additional benefits in enhancing the persons' feeling of control at a time when they have experienced the loss of their ability to void naturally and when their psychological resilience might, in consequence, be low.

Thank you for these comments regarding the paper.

Reviewer 2: Maria Hälleberg Nyman

This is an important study on re-use of intermittent catheters from the users' perspective.

Thank you for your comments about the importance of this study from the users' perspective

1. Is the research question or study objective clearly defined? Yes

For clarity, we have made sure that the objective stated in the abstract is the same as the 'purpose of the study' stated at the end of the introduction (see pages 2 and 4).

2. Is the abstract accurate, balanced and complete?

Setting should be 12 general practice clinics in Hampshire and Dorset, UK.

We have left the 'setting' in the abstract as 'participant's own homes', as all of the qualitative interviews took place in the participant's own homes. The role of the GP surgeries was to identify the patients and send the invitation letters only; the interviews did not take place in the GP setting (see page 2).

The abstract might need to be re-written if the result section is rewritten.

We have now re-written the abstract 'results section' so that the 4 main themes are clearly stated at the beginning of the 'results' section and the findings described in the abstract follow the order of the 4 themes. In the original submission the themes were described in a different order. We hope this makes it clearer (see page 2).

3. Is the study design appropriate to answer the research question? Yes

4. Are the methods described sufficiently to allow the study to be repeated?

The data analysis is not described in detail. And there is no methodological reference for the thematic analysis. Is the analysis performed in the 6 steps described by Braun and Clarke?

The thematic analysis was based on the method described by Braun and Clarke, so we have now referenced Braun and Clarke in that section and added a sentence to describe the process further (see page 5). We used a similar description of the analysis method in a previous paper, which was part of the wider MultiCath programme using the same participants, submitted to BMJopen (reference: Okamoto, I et al (2017) Intermittent catheter users' symptom identification, description and management of urinary tract infection: a qualitative study BMJopen; 7:e16453).

5. Are research ethics (e.g. participant consent, ethics approval) addressed appropriately? Yes

6. Are the outcomes clearly defined? Yes

7. If statistics are used are they appropriate and described fully? NA

8. Are the references up-to-date and appropriate? Yes

9. Do the results address the research question or objective? Yes

10. Are they presented clearly?

The result section is a bit hard to follow. There are four themes and eleven sub-themes. The result seems to be based on raw data rather than analyzed material. The authors provide a lot of quotations but not so much text to describe the themes and sub-themes. This indicates that data should be analyzed further.

We can see that this was an issue to be addressed and have revised the results section. We have looked further at the codes, sub-themes and themes and more description is included and fewer quotations in the results section of the paper. We have amended Figure 1 so that it now includes the main themes and sub-themes and the codes from which these were derived. We hope this shows the analysis and findings more clearly (see pages 5-11).

Beside the themes and sub-themes the authors provide a figure with factors affecting acceptability of re-use – some of these factors are the same as a theme or sub-theme and some are not. This is very confusing.

We agree that the original Figure 1 does not include all the themes and sub-themes that came from the analysis, and therefore may not be clear. We have removed this version of Figure 1 and have replaced it with a new version, which shows the main themes and sub-themes and these correspond to all of the themes described in the results section. We hope this makes the results section more coherent and clearer and that this visual representation now aids the reader.

Also, the authors provide characteristics for the participants that re-use catheters but not for those that do not re-use catheters.

We have included more details about the participants that re-use because they were able to discuss their actual experiences of re-use in the interviews, but this is now included in a paragraph rather than a table (see page 5).

11. Are the discussion and conclusions justified by the results?

The result section needs to be re-written, as a result of this discussion and conclusions also might be

re-written.

We have made the themes and sub-themes clearer in the results section, by referring to them from the beginning of that section and providing a diagram of all the themes and sub-themes. The discussion had slightly different paragraph headings to those of the 4 main themes and we have now amended them to ensure they are in line with the main themes (see pages 11-13).

12. Are the study limitations discussed adequately?

The authors discuss a few limitations, but they do not discuss the fact that the interview-guide include some leading questions e.g. the questions on the last page of the interview-guide.

The inclusion of questions about the advantages and disadvantages of both methods may have influenced participant's responses, but the questions were asked in the same way for single-use and re-use of catheters. We have added this to the limitations paragraph (see page 13).

13. Is the supplementary reporting complete (e.g. trial registration; funding details; CONSORT, STROBE or PRISMA checklist)?

Funding details are provided, but no checklist e.g COREQ

We have now attached the COREQ checklist.

14. To the best of your knowledge is the paper free from concerns over publication ethics (e.g. plagiarism, redundant publication, undeclared conflicts of interest)? Yes

15. Is the standard of written English acceptable for publication? Yes

We hope that we have addressed all the issues raised by the editor and reviewers in the revised version of the manuscript and await further comments from the editor regarding this submission.